# *Comet:* A COMMUNICATION-EFFICIENT AND PERFORMANT APPROXIMATION FOR PRIVATE TRANSFORMER INFERENCE

## ABSTRACT

The prevalent use of Transformer-like models, exemplified by ChatGPT in modern language processing applications, underscores the critical need for enabling private inference essential for many cloud-based services reliant on such models. However, current privacy-preserving frameworks impose significant communication burden, especially for non-linear computation in Transformer model. In this paper, we introduce a novel plug-in method *Comet* to effectively reduce the communication cost without compromising the inference performance. We second introduce an efficient approximation method to eliminate the heavy communication in finding good initial approximation. We evaluate our *Comet* on Bert and RoBERTa models with GLUE benchmark datasets, showing up to $3.9\times$ less communication and $3.5\times$ speedups while keep competitive model performance compared to the prior art.

## 1 INTRODUCTION

Leveraging the Transformer-based architecture (Vaswani et al. (2017)), the Generative Pretrained Transformer (GPT) (Brown et al. (2020)) is reshaping the global deep learning applications (Wang et al. (2022; 2024)) landscape by demonstrating remarkable proficiency in comprehending human language and generating multifaceted content. For instance, users can receive instructional responses by sending queries via ChatGPT web portal. While such client-server interaction scheme enhances efficiency and productivity, privacy has emerged as a concern. More specifically, machine learning applications like ChatGPT require either users provide language prompts or images, which may include confidential information, to the service provider. On the other hand, the server has concerns on exposing trained model weights, which are considered as vital asset, to the clients. Therefore, the gap between privacy requirements and efficient performance motivates our study of private Transformer inference.

To address the privacy concerns (Garcia et al. (2023)) of users and protect the model on the server, several privacy-preserving inference frameworks (Rathee et al. (2020); Mishra et al. (2020); Xu et al. (2024)) have been proposed for convolutional neural networks via applying secure multi-party computation (MPC) techniques, such as homomorphic encryption (HE) (Fan & Vercauteren (2012); Cheon et al. (2017)), secret sharing (SS) (Shamir (1979)), and oblivious transfer (OT) (Brassard et al. (1987)). However, directly applying such privacy-preserving frameworks to Transformers leads to overwhelming computing and communication cost, because *the Transformer-based models usually face complex hybrid protocols for non-linear functions like GeLU, Softmax, and LayerNorm*, which have not been sufficiently addressed in previous studies. To facilitate the widespread of private Transformer inference services, several works (Hao et al. (2022); Li et al. (2022)) propose customized protocols and fine-tuning model for reducing communication cost. However, these existing works still encounter *the challenge of heavy communication required to find good initial approximations or lengthy fine-tuning processes*. For example, our investigations have shown that the Look-Up Table (LUT) method, extensively applied in the state-of-the-art work Iron, necessitates heavy communication on capturing good initial approximations (Rathee et al. (2021); Hao et al. (2022)). To alleviate such communication burdens, MPCFormer (Li et al. (2022)) replaces two heavy non-linear functions, namely Softmax and GeLU, with aggressive quadratic polynomials for communication reduction, albeit at the cost of requireing further lengthy fine-tuning and compromises to lower performance. Based on our preliminary explorations, we gain observations about the empirical

| LUT | Polynomial (piecewise) |
|---|---|
| Hao et al. (2022); Huang et al. (2022) | Fan et al. Pang et al. (2024) |
| Rathee et al. (2021) | Lu et al. (2023) Luo et al. (2024) Dong et al. (2023) |

Table 1: Existing works taxonomy of non-linear functions for private inference

characteristics of Transformer model and secret sharing techniques. These findings present us opportunities for designing of communication-efficient and performant private Transformer inference:

*Smoothed approximation for Transformer inference:* We observe that smoothed approximation functions can maintain or even enhance the Transformer performance across various tasks when replacing GeLU activation function. Additionally, our experimental results indicate that Softmax function replaced by $\frac{ReLU(x)}{\sum ReLU(x)}$ has marginal influence on model accuracy, which echos the finding of secureML (Mohassel & Zhang (2017)). This provides us an opportunity to unify common non-linear protocols to one function through the inverse square root.

*Affinity of exponent of secret shares:* Current protocols on calculating the inverse square root involve high communication costs, often utilizing LUT or (piecewise) polynomial approximation to find good initial approximations for consecutive Newton iterations. We discover that such heavy communication can be totally removed via our novel design protocol. Because the magnitude of activation values is around zero, this provides us a unique opportunity to propose a share flooding technique to ensure our novel protocol working securely in two-party mode.

Based on the above observations, we propose *Comet*, a communication-friendly and performant private Transformer plug-in approximation method. *Comet* unifies hybrid complex non-linear functions and designs new specialized protocols to eliminate most of the communication for unified non-linear function. Specifically, in Sec. 3.1, we first endeavor to harmonize non-linear functions that applies hybrid complex protocols, namely GeLU and Softmax, with smoothed maximum unit (SMU) function (Biswas et al. (2022)). In Sec. 3.2, we present our novel double approximation protocol that removes the communication cost of finding the initial approximation when calculating the inverse square root. To facilitate the proposed double approximation protocol applied in two-party computation scheme, we design a share flooding technique to render the method fully practical, thereby avoiding potential divergence after Newton iterations in Sec. 3.3. We implement our method and conduct extensive evaluations with BERT (Kenton & Toutanova (2019)) and RoBERTa-base (Liu et al. (2019)) models on the GLUE benchmark (Wang et al. (2018)) in Sec. 4. Our experiment results show that *Comet* achieves up to $3.9\times$ reduction in communication cost and $3.5\times$ time speedup, compared with LUT method and common Taylor approximation method utilizing the state-of-the-art framework Iron (Hao et al. (2022)) and CrypTen (Knott et al. (2021)), respectively.

## 2 PRELIMINARIES

### 2.1 THREAT MODEL

Similar to previous works (Juvekar et al. (2018); Riazi et al. (2019); Hao et al. (2022)), our method follows the *two-party semi-honest* threat model. Specifically, the client $C$ and the server $S$ follow the protocol but attempt to infer each other's input, namely the client's input data and the server's model parameters, during the inference process.

### 2.2 ADDITIVE SECRET SHARING AND PROTOCOLS

Given an original message $m$ at party $P \in \{0, 1\}$, one of the two Additive Secret Shares (ASS) is constructed by uniformly sampling randomness $r$ and setting $\langle m \rangle_P = r$, while the other share is formed as $\langle m \rangle_{1-P} = m - r$. To reconstruct the message, one can simply add two shares $m = \langle m \rangle_P + \langle m \rangle_{1-P}$. In this work, we utilize ASS to share the encrypted output of linear functions. Existing research has developed accurate computation protocols for non-linear function using secret sharing, which protect the privacy of both the client and server. SIRNN (Rathee et al. (2021)) designs multiple accurate non-linear computation protocols for convolution neural network, extensively using LUT for layer normalization, Softmax, and exponential function. The functionality of LUT takes

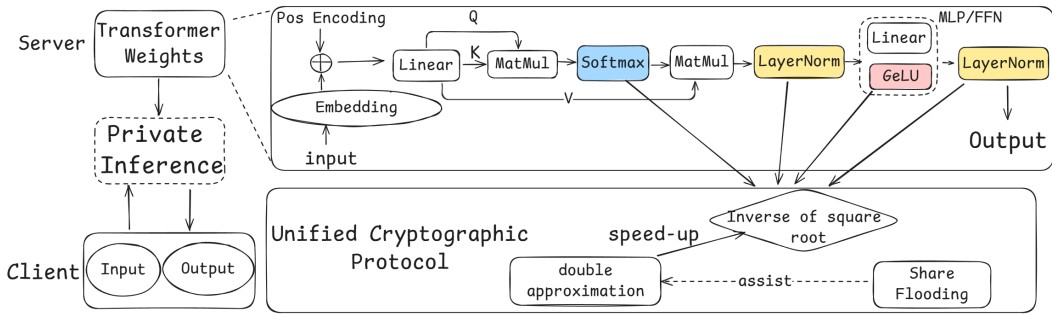

Figure 1: The overview of *Comet*(The figure is updated.). The client (left), who holds the input, interacts with and receives output from the server, who holds the model, via private Inference engines, e.g., CrypTen and Iron. *Comet* (right) unifies the non-linear functions into inverse square root and save communication with double approximation and share flooding in two-party mode.

as input string $i$ (with $\sigma$ bitwidth) and output $T < i >$ (with $\alpha$ bitwidth) where $T$ is a M-entries table. Such functionality can be achieved via $\binom{M}{1}$-OT with $(2^\sigma - \log M)$ offline and $(M * \alpha + \sigma)$ online communication bits (Brüggemann et al.; Ito et al. (1997)). Iron (Hao et al. (2022)) improves the communication efficiency of relevant protocols via customized optimizations. Despite efforts to reduce communication cost regarding accurate protocols, current protocols continue to face significant communication burdens. To ease such overhead, (piecewise) polynomial approximation (e.g., Taylor expansion) can greatly reduce the communication overhead by converting complex protocols to multiplications (Fan et al.; Chou et al. (2018)). Nevertheless, the use of low-degree polynomials leads to a notable loss in inference performance, while employing high-degree polynomials incurs substantial communication costs.

## 2.3 IEEE 754 FLOAT-POINT REPRESENTATION

The IEEE 754 is a technical standard for floating-point representation (Kahan (1996)). Any floating-point number can be represented in the form of $(1+m)*2^e$, where $m$ is mantissa ($m \in [0, 1)$) and $e$ is exponent number. To store a floating-point number in IEEE 754 representation, taking 32-bit floating-point number as example, 3 basic components should be filled: The highest bit denotes the sign of floating-point number, where 0 represents a positive number while 1 represents a negative number; The exponent (E), as shown in Fig. 2 (green part), is filled by adding a bias B$= 2^7 - 1$ to the actual exponent (e), which means $E = e + B$. The Normalised Mantissa (m) can be directly filled in the binary form as shown in red part of Fig. 2. For convenience, we denote M as the integer mantissa,

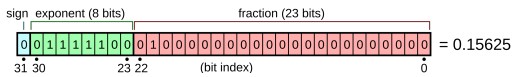

Figure 2: a IEEE 754 Floating-point representation example

which is generated by moving fraction dot to the end of mantissa ($M = L * m$), where $L = 2^{23}$. $M + EL$ denotes the binary content corresponding to a floating-point number.

## 2.4 NEWTON-RAPHSON METHOD

The Newton-Raphson method (Ramamoorthy et al. (1972)) is a root-finding algorithm which produces successively better approximations to the roots of a real-valued function. Given a single-variable function $f$ defined for a real variable x, the derivative $f'$, and an initial guess $x_0$ for a root of $f$, if the function satisfies sufficient assumptions and the initial guess is close, then $x_1 = x_0 - \frac{f(x_0)}{f'(x_0)}$ is a better approximation of the root than $x_0$. Geometrically, $(x_1, 0)$ is the intersection of the x-axis and the tangent of the graph of $f$ at $(x_0, f(x_0))$), or in other words, the improved guess is the unique root of the linear approximation at the initial point. The process is repeated as $x_{n+1} = x_n - \frac{f(x_n)}{f'(x_n)}$ until a sufficiently precise value or a predefined number of iterations is reached. In this study, we follow the Newton Iteration equation $x_{n+1} = x_n(\frac{3}{2} - \frac{x}{2}x_n^2)$ for inverse square root in previous works (Knott

et al. (2021); Ramamoorthy et al. (1972); James & Jarratt (1965); Schulte & Wires (1999)), where $x$ is the input and $x_n$ is the estimation result for each iteration.

## 2.5 PRIVATE TRANSFORMER INFERENCE

*Comet*, like previous related works (Rathee et al. (2020); Hao et al. (2022)), considers the scenarios where the server holds Transformer model, while the client holds and sends private input, as shown in the left part of Fig. 1. Our framework enables clients sending inference requests and receives prediction results on its input. To keep the privacy of both client and server, several private Transformer inference frameworks (Hao et al. (2022); Li et al. (2022)) are introduced. These frameworks mainly divide the cryptographic operations into two categories– linear and non-linear. For linear computation like matrix multiplication, HE is commonly used. HE ciphertext allows operations like multiplication on ciphertext without decryption. To maintain the correctness of decryption, ciphertexts need to be refreshed within limited number of operations via bootstrapping (Chillotti et al. (2016)) or re-encryption. To produce the linear results of each attention layers, the client encrypts their input embedding vectors, with coefficient encoding technique (Hao et al. (2022); Huang et al. (2022)), into HE ciphertexts and send them to the server. The server multiplies the ciphertexts-plaintext matrix with HE multiplication. To protect the privacy of kernel parameters, the server should generate two shares with random number mask and send one encrypted share to the client for consecutive non-linear functions.

GeLU, LayerNorm, and Softmax are the most common non-linear functions in Transformer-like models, which induce over 80% of communication cost in inference (Li et al. (2022)). For example, GeLU necessitates the Gaussian Error function $erf(x)$, which is approximated using a high-order Taylor expansion. This introduces much more rounds of multiplications compared with linear part of private Transformer inference. Similar challenges are faced in Softmax and LayerNorm, which require to calculate exponential function and inverse square root. Furthermore, the variety of non-linear functions imposes extra difficulties for reducing communication cost, since different customized protocols must be developed for each non-linear function, such as exponential and complex $erf$ function. Current works apply high-order polynomial approximation to estimate such complex function in (Lu et al. (2023); Pang et al. (2024); Dong et al. (2023); Luo et al. (2024)), compromising to model performance or inference latency.

## 3 METHOD

In this section, we present *Comet*, which unifies the hybrid complex non-linear protocols and removes the significant communication cost for finding good initial approximation. We provide details on how to unify protocols in Sec. 3.1, how to transfer communication to local computation in Sec. 3.2, and how to avoid divergence with share flooding in Sec. 3.3.

### 3.1 UNIFY HYBRID COMPLEX PROTOCOLS

To address the heavy communication issue in private Transformer inference, we first shed light on the Transformer architecture, which consists of multiple encoder-decoder architecture. The encoder has similar structure with the decoder, hence we focus on encoder blocks. A typical Transformer model with multiple encoder blocks consists of (1) an embedding layer, (2) a stack of encoder blocks, and (3) a prediction layer. One input token maps to a latent representation vector via the embedding layer. The encoder blocks are composed of attention layers and feed-forward layers as illustrated below.

**Attention layers.** After taking in the token embedding vector, attention function attempts to generate query, key, and values vectors with corresponding weights, denotes as $X_Q$, $X_K$, and $X_V$, respectively. Then, the layer output the attention vector using the function: $Attention(X_Q, X_K, X_V) = Softmax(\frac{X_Q X_K^T}{\sqrt{d}})X_V$, where $d$ is the dimension of embedding vector.

**Feed-forward Layers.** The layer can be represented as follows: $FeedForward(X) = GeLU(XW_1 + b_1)W_2 + b_2$, where GeLU is the Gaussian Error Linear Unit function. The feed-forward layer takes the output attention vectors of attention layers as input. The GeLU function requires Gaussian Error function $erf(x)$: $GeLU(x) = 0.5x * (1 + erf(\frac{x}{\sqrt{2}}))$.

**Layer Normalization.** Layer Normalization (LayerNorm) is applied after attention layers and feed-forward layers. By calculating the normalization of mini-batch of input, it smooths gradients for better generalization accuracy, shown in following equation: $y = \frac{x - E(x)}{\sqrt{Var(x) + \epsilon}} * \gamma + \beta$ , where $E(x)$ denotes the mean of input $x$ and $Var(x)$ is the variance of input, $\gamma$ and $\beta$ are learnable parameter during the training.

$e^x$ in Softmax, $erf(x)$ in GeLU, and $\frac{1}{\sqrt{x}}$ in LayerNorm require either different Taylor approximation functions or different specialized LUT protocols to calculate their results. To unify the complex protocols in non-linear functions, we propose to replace the exponential function $e^x$ in Softmax with ReLU function as SecureML (Mohassel & Zhang (2017)), i.e., $Softmax^*(x) = \frac{ReLU(x)}{\sum ReLU(x)}$ since it shows competent performance in experiments (see in Sec. 4.2 and Appendix. C). Since the erf function in $GeLU$ requires high order Taylor expansion in specialized protocol, we leverage the smoothed maximum unit (SMU) to replace the GeLU function. The SMU function for GeLU derives as $smu(x) = \frac{1+\alpha}{2} * x + \frac{(1-\alpha)*x^2 + \mu^2}{2*\sqrt{(1-\alpha)*x^2 + \mu^2}}$, where $\alpha$ and $\mu$ are the trainable parameters to control the slope of negative axis and smoothness of function, respectively (refer to Appendix. B for detailed derivation and Appendix. F for explanation). We set $\alpha = 0, \mu = 0$ for ReLU replacement in $Softmax^*$ function, and $\alpha = 0, \mu = \frac{1}{\sqrt{2}}$ for GeLU replacement in $smu(x)$ function to retain the model performance. Additionally, we test the flexibility of $\alpha$ and $\mu$ by training from scratch, showing a performance boost in some tasks of GLUE benchmark in Appendix. C. As the LayerNorm only requires inverse square root, we successfully unify all non-linear functions, namely GeLU, Softmax and LayerNorm, into inverse square root.

## 3.2 Double Approximation

Even though we unify complex non-linear protocols to the inverse square root, we still face the high communication challenge when calculating this function. Current works on calculating the inverse square root requires the Newton iterations, or Goldschmidt's iteration (Ercegovac et al. (2000)), to get accurate results with a initial approximation. Due to the local convergence characteristic of Newton method (refer to Appendix. E for details), the initial approximation has to be close enough to the root of function, referred as "the good initial approximation". However, finding the good initial approximation usually requires over $85\%$ communications of total process using LUT method or high order Taylor expansion (Rathee et al. (2021)). In this manner, we demonstrate our double approximation method for such initial approximation finding without communication for inverse square root $y = \frac{1}{\sqrt{x}}$. First, we take logarithm on both sides of the equation to $log(y) = -\frac{1}{2}log(x)$. Such equation can be easily transformed into secret-shared form as equation ( 1), where $x_c$ and $x_s$ denote as the share of client and server.

$$
\begin{aligned}
2log(y) &= -log(x) \\
&= -log(x_c + x_s)
\end{aligned}
\tag{1}
$$

Then we replaced the input x and output y with IEEE 754 floating-point representation:

$$
\begin{aligned}
2log((1+m_y) * 2^{e_y}) &= -log((1+m_x) * 2^{e_x}) \\
&= -log((1+m_{x_c}) * 2^{e_{x_c}}) + (1+m_{x_s}) * 2^{e_{x_s}}))
\end{aligned}
\tag{2}
$$

First, let us focus on one-party calculation of inverse square root, which is the upper equation in equation (2). We change the multiplication in logarithm to addition following the logarithm rule.

$$
2(log(1+m_y) + e_y) = -log(1+m_x) + e_x
\tag{3}
$$

As the logarithm can be complex to calculate, we first approximate the logarithm with linear function $log(1+m) \approx m + b, m \in [0, 1)$, where $b$ is a constant number we can predefined based on logarithm function. After the replacement of logarithm with **first approximation**, we have the equation (4) and reorganized to equation (5) to estimate the inverse square root value:

$$
2(m_y + b + e_y) \approx -(m_x + b + e_x)
\tag{4}
$$

$$
(m_y + e_y) \approx -(m_x + e_x)/2 - 3b/2
\tag{5}
$$

Such approximation can be efficient to calculate the inverse square root with good precision in one-party mode. However, it is challenging for the secret share scheme, as the logarithm cannot be replaced when two shares are added in the lower part of equation (2). Our insight is that two shares' exponent part can be approximately equal to each other, as the exponent parts for two shares are only 8-bits length. Then we take **the second approximation** $e_{x_c} \approx e_{x_s}$ as hypothesis and satisfied in Sec. 3.3. We can further transform the secret-shared form equation (2) with second approximation as following:

$$2log((1 + m_y) * 2^{e_y}) \approx -log((1 + m_{x_c} + 1 + m_{x_s}) * 2^{e_{x_s}})) \tag{6}$$

$$2(log(1 + m_y) + e_y) \approx -log((1 + (m_{x_c} + m_{x_s})/2) * 2^{e_{x_s}+1}))$$
$$= -log(1 + (m_{x_c} + m_{x_s})/2) - e_{x_s} - 1 \tag{7}$$

Then we apply the first approximation to the lower equation (7):

$$2(m_y + b + e_y) \approx -(m_{x_c} + m_{x_s})/2 - b - e_{x_s} - 1 \tag{8}$$

We then replace the $m$ and $e$ with $M = L * m$ and $E = e + B$ as stated in Sec. 2.3:

$$2\frac{M_y}{L} + 3b + 2(E_y - B) \approx -\frac{(M_{x_c} + M_{x_s})}{2L} - E_{x_s} + B - 1 \tag{9}$$

We reorganize (9) to the equation (10) into two shares mode, where $E_{x_{c|s}}$ denotes we replace the $E_{x_c}$ with $E_{x_s}$ as second approximation:

$$M_y + LE_y \approx \boxed{-\frac{1}{4}(M_{x_c} + LE_{x_c})}_c \boxed{-\frac{1}{4}(M_{x_s} + LE_{x_{c|s}})}_s \boxed{+\frac{(3B - 3b - 1)L}{2}}_s \tag{10}$$

The orange and blue term can be regarded as the integer value of the client and the server share, respectively. The last black term is a constant that both parties can learn offline. The boxed term with undertext $c$ and $s$ are the output shares of approximated inverse square root result for client and server, respectively.

In this manner, we can get a approximated value of inverse square root without heavy communication between the client and server, as the client and server can only calculate on their own shares and a constant. The initial approximation can be produced by adding two shares. The precise result can be approached via Newton Method with 3-4 iterations, as shown in Sec. 4.3. As the main bottleneck of communication lies in finding good initial approximation, we decrease $\mathcal{O}(2^\sigma)$ LUT communication cost to $\mathcal{O}(1)$ in private Transformer inference.

### 3.3 SHARE FLOODING

In two-party mode, server needs to generate the shares for client and server itself with random number mask, after the linear results are produced in HE ciphertext. However, our second approximation requires the exponent part of two shares are close to each other to remove the communication cost. If the exponent part is not close enough, the result of Newton iteration would be diverge (see Appendix D for more detailed experiments). As shown in Fig. 3, the random number in the gray box which is too close to the input values can make the counterpart share far from each other, e.g., if the input value is 3.1 and random number is 3.098, the other share 0.002 would generate bad initial approximation that leads to divergence, since the large difference in the exponent of two shares breaks second approximation. This imposes a dilemma for generating two shares as one of the share is the uniformly random number mask that is generated offline by server. It is infeasible to always meet the requirements of second approximation as the server cannot learn the convolution results before generating the random number mask offline. This dilemma brings us a new challenge— *How can we securely generate shares and efficiently perform double approximation while satisfying privacy needs and approximation assumptions?*

To address this challenge, we propose the share flooding technique for private Transformer inference. Our insight is that the absolute magnitude of tensor values is closely surrounded around zero. This is because the input embedding vector is a 0 to 1 valued vector after softmax using word2vec technique (Mikolov et al. (2013)). Applying Softmax and LayerNorm function in attention block and feed-forward layers iterative also confine the activation magnitude into zero to one range. Meanwhile, the

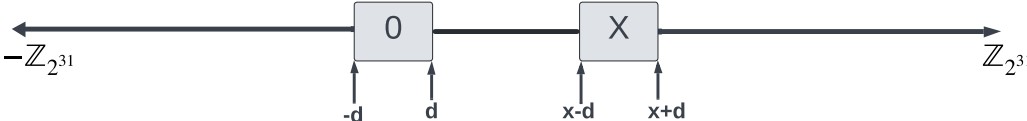

Figure 3: Demonstration of double approximation divergence example. "d" denotes the upper bound of exponent between two shares that would lead to divergence in Newton Method. One share generated, in integer field $\mathbb{Z}_{2^{31}}$, in grey box (within the bound d) means the share is too close to the input X, which makes the two shares exponent out of the convergence bound of Newton Method.

training process usually applies L1/L2 normalization in loss function to discourage large weights to fight overfitting issue, which makes the consecutive activation magnitude to be small as well. This is also validated in our preliminary experiments. We refer readers to Appendix A for more details.

With our insight, we design to flood the random number mask with a large absolute value. This flood number can drown the exponent of two shares to satisfy the second approximation requirement. For example, if the flood number is 8192 and input message is 3.1, we add the flood number to the random number mask, such as 3.098, results in 8195.098. The other share is -8191.998. Both shares have the same exponent equal to 140 in IEEE 754 float-point representation. To be more precise, we flood the random mask offline with a large adjustable flooding number to specific task, as one party's share. The corresponding other share is generated when online message subtract with flooded random mask, then transfer the floating-point valued share to the integers in corresponding integer field. As equation (10) shows, the flooding number cannot be offset as the blue and orange shares are same sign. We compensate the over-flooded value by adding $\frac{1}{2}(E_f - E_m)L$, where $E_f$ is the exponent of flooding number and $E_m$ denotes the exponent of most frequent activation value of the distribution learned in similar tasks. Note that the fixed-point shares can be easily transfer to floating-point shares by dividing the scale (see details in Appendix G). We securely produce shares as there is no information exposed except the sign of two shares, which would not expose any information of original input value. This novel design enable us for addressing the challenge of security and approximation assumption requirement.

## 4 EXPERIMENTS

### 4.1 EXPERIMENTAL SETTINGS

We implement *Comet* within the secure two-party framework Iron, which uses the EMP toolkit (emp) for implementing non-linear functions, and the CrypTen framework. The experiments are conduct on two servers with an AMD EPYC 7413 24-core Processor 64GB RAM, under the network bandwidth of 200Mbps. We set the flooding number equal to 8192 ($E_f = 140, M_f = 0$), as it only floods the exponent $E_f$ with mantissa $M_f$ equals to zero, and $E_m = 128$, as it covers most of the activation distribution in Transformer-based model. We evaluate the model performance based on HuggingFace implementation with the dataset of GLUE benchmarks. In following sections, we aim to answer three questions to present the benefits of *Comet*: (1) The model performance of the unified model. (2) The iteration number of the Newton method required and its effect on model performance. (3) The time and communication reduction of *Comet*.

### 4.2 UNIFIED MODEL PERFORMANCE

To present the model performance of the unified model, we evaluate its performance use the Bert-base and RoBERTa-base model and compare it with baseline models. We use the sequence length of 128 for the selected datasets of GLUE benchmark. Baseline models are selected combinations within the set of {GeLU, s-GeLU, Softmax, $Softmax^*$, $s - Softmax^*$ }, where "s-" denotes the $smu(x)$-replaced ReLU or GeLU in the following function. We train the baseline models with learning rate from 1e-6, 5e-6, 1e-5, and 1e-4, the number of epochs from 10, 30, and 100. We also set the $\alpha = 0, \mu = 0$ for $s - Softmax^*$ and $\alpha = 0, \mu = \frac{1}{\sqrt{2}}$ for s-GeLU, as stated in Sec. 3.1.

Table 2: The model performance of a subset of GLUE benchmark with different combination of smoothed maximum unit replacement for GeLU and Softmax function. "$Softmax^*$" stands for ReLU replaced Softmax in Sec. 3.1. "s-" stands for the $smu(x)$ smoothed function in Sec. 3.1. Average Pearson and Spearman correlation is reported for STS-B. Matthews correlation is reported for CoLA. Accuracy is reported for other datasets.

| Bert-base | RTE | MRPC | STS-B | SST-2 | CoLA |
|---|---|---|---|---|---|
| GeLU + Softmax | 70.8 | 88.97 | 88.6 | 92.7 | 58.9 |
| GeLU + $Softmax^*$ | 66.4 | 86.9 | 85.8 | 91.4 | 54.2 |
| GeLU + s-$Softmax^*$ | 67.9 | 86.3 | 86.7 | 90.5 | 55.7 |
| s-GeLU + Softmax | 69.4 | 86.7 | 88.3 | 91.1 | 56.1 |
| s-GeLU + $Softmax^*$ | 67.2 | 88.1 | 88.1 | 92.1 | 55.7 |
| s-GeLU + s-$Softmax^*$ | 71.5 | 88.6 | 88.8 | 92.6 | 57.9 |
| RoBERTa-base | | | | | |
| GeLU + Softmax | 74.2 | 92.3 | 89.1 | 92.1 | 55.7 |
| GeLU + $Softmax^*$ | 71.8 | 89.6 | 84.8 | 89.9 | 53.8 |
| GeLU + s-$Softmax^*$ | 72.6 | 88.9 | 87.2 | 89.1 | 52.9 |
| s-GeLU + Softmax | 72.7 | 89.1 | 88.1 | 87.3 | 54.2 |
| s-GeLU + $Softmax^*$ | 73.4 | 91.7 | 86.3 | 88.4 | 54.1 |
| s-GeLU + s-$Softmax^*$ | 74.9 | 92.2 | 90.1 | 90.4 | 55.8 |

Table. 2 upper part shows the model performance of Bert model on a subset of GLUE benchmark with different baseline models. Since all non-linear functions are unified to inverse square root with SMU unit, we name our target model with "s-GeLU + s-$Softmax^*$", as the unified model. The unified model achieves marginal performance loss with less than $1\%$. It also shows a small accuracy boost of approximately $1\%$ in the relatively small dataset RTE, while other baselines experience larger performance loss. In this manner, our unified model preserves model performance with Bert-base model. To validate our observation, we also evaluate the unified model with RoBERTa-base model architecture with same datasets. As shown in lower part of Table. 2, the target unified model of RoBERTa-base shows consistent model performance among the GLUE benchmark subset.

### 4.3 NEWTON ITERATION EVALUATION ON MODEL PERFORMANCE

In this section, we evaluate the double approximation method of *Comet* in model performance. To answer the question of how many iterations that the double approximation method requires to recover the performance of unified model, we conduct experiments on the unified model with varying iteration number when calculating the inverse square root with double approximation method. We set $b = 0.045$ by inversely solving $\frac{3}{2}L(B - b) = $ 0x5f3759df derived from equation (5) with $m = M/L$ and $e = E - B$ replaced, given the magic number 0x5f3759df as in fast inverse square root algorithm (Lomont (2003)). Such magic number can be obtained by minimizing the relative error between the approximated results and real results as shown in (Lomont (2003)). Our double approximation method can recover the model performance of the unified model in 3~4 iterations with the initial approximation resulting from our method, as shown in Table. 3. Our method requires fewer iterations compared to the CrypTen, where the inverse square root requires 10 iterations by default with a communication-intensive initial approximation function of $2.2 * e^{(-0.5x+0.2)} + 0.2$. Even though we incur around extra 2 rounds compared with LUT method (which only requires 1~2 rounds), LUT method necessitates heavy communication for accurate initial approximation, and such communication can grow exponentially with the number of table entries. Thus, our method outperforms the LUT in the total communication and we elaborate the details in Sec. 4.4.

### 4.4 END-TO-END INFERENCE COMMUNICATION AND TIME COMPARISON

In this section, we evaluate the end-to-end inference time and communication cost of *Comet* implemented within the state-of-the-art privacy-preserving frameworks Iron and CrypTen. We first compare our method with the $2.2 * e^{(-0.5x+0.2)} + 0.2$ function applied in CrypTen and high/low order of Taylor expansion for finding initial approximation of inverse square root in both communication and time using the unified model. *Comet* recovers the model performance with setting of 4 Newton

Table 3: The model performance of double approximation method on how many iterations to recover the best model performance from initial approximation. The unmodified model is the model with unchanged GeLU and Softmax functions. The unified model is the SMU-replaced GeLU and SMU-replaced $Softmax^*$ function model.

| Bert-base | RTE | MRPC | STS-B | SST-2 | CoLA |
|---|---|---|---|---|---|
| Unmodified Model | 70.8 | 88.97 | 88.6 | 92.7 | 58.9 |
| Unified Model | 71.5 | 88.6 | 88.8 | 92.6 | 57.9 |
| Without Newton Iteration | 64.6 | 81.58 | 86.7 | 90.8 | 52.6 |
| 3 iterations | 70.8 | 86.9 | 88.6 | 92.2 | 56.4 |
| 4 iterations | 71.5 | 88.6 | 87.6 | 92.5 | 57.8 |
| RoBERTa-base | | | | | |
| Unmodified Model | 74.2 | 92.3 | 89.1 | 92.1 | 55.7 |
| Unified Model | 74.9 | 92.2 | 90.1 | 90.4 | 55.8 |
| Without Newton Iteration | 66.7 | 87.8 | 86.7 | 85.3 | 51.9 |
| 3 iterations | 72.4 | 86.7 | 90.1 | 89.4 | 53.9 |
| 4 iterations | 74.2 | 88.1 | 92.1 | 90.6 | 55.7 |

iterations as shown in Sec. 4.3, while high and low order Taylor approximation would take 2 and 8 iterations, respectively. We can see from Table. 4 that *Comet* achieves a 2.5× speedup compared to the original CrypTen framework, with 2× to 3× communication reduction for different non-linear functions. Such improvement is attributed to the removal of the bottleneck communication involved in finding initial approximation. We also compare *Comet* with Taylor approximation with high/low order polynomial, namely $1 - \frac{x}{2} + \frac{1}{2} + \frac{3(x-1)^2}{8} - \frac{5(x-1)^3}{16} + \frac{35(x-1)^4}{128} - \frac{63(x-1)^5}{256} + \frac{231(x-1)^6}{1024} - \frac{429(x-1)^7}{2048}$ and $1 - \frac{x}{2} + \frac{1}{2} + \frac{3(x-1)^2}{8}$, showing similar $4.0\times \sim 2.8\times$ reduction in time and $3.2\times \sim 1.9\times$ in communication. This is because the high-order Taylor approximation requires significant communication for calculating the initial approximation, whereas the low-order Taylor approximation typically demands more iterations and may diverge due to the initial approximation surpassing the requirement for local convergence of the Newton Method.

Table 4: The communication (GB) and inference time (Second) comparison on Bert-base and RoBERTa-base model with CrypTen. "H-Taylor P" and "L-Taylor P" denote 7 order Taylor polynomial and 2 order Taylor polynomial approximating inverse square root generated at 1, respectively.

| Bert-base | Total Time(s) | LayerNorm Time(s) | LayerNorm Comm | Act Time | Act Comm | Softmax Time | Softmax Comm |
|---|---|---|---|---|---|---|---|
| CrypTen | 74.8 | 12.19 | 2.14 | 34.1 | 6.28 | 17.8 | 3.28 |
| H-Taylor P | 73.2 | 13.6 | 2.33 | 32.6 | 5.82 | 16.2 | 3.19 |
| L-Taylor P | 70.6 | 10.5 | 2.03 | 29.6 | 5.57 | 15.8 | 3.09 |
| *Comet* | **29.8(2.5×)** | **3.32(3.7×)** | **0.94(2.3×)** | **7.92(4.3×)** | **1.83(3.4×)** | **5.7(3.1×)** | **1.58(2.07×)** |
| RoBERTa | | | | | | | |
| CrypTen | 79.3 | 14.19 | 2.6 | 36.3 | 7.4 | 18.2 | 3.34 |
| H-Taylor P | 75.7 | 13.9 | 2.32 | 34.7 | 7.31 | 17.6 | 3.19 |
| L-Taylor P | 73.1 | 13.3 | 2.21 | 30.9 | 6.4 | 15.1 | 3.1 |
| *Comet* | **34.8(2.3×)** | **5.81(2.4×)** | **1.59(1.6×)** | **9.1(4.0×)** | **1.88(3.9×)** | **6.9(2.63×)** | **1.79(1.8×)** |

We also compare *Comet* to LUT-based framework Iron in the same experimental setting as we did with CrypTen. Our method achieves up to 3.48× for time speedup and 2.4× to 3.7× communication reduction in non-linear functions, as shown in Table. 5. This runtime improvement and communication reduction demonstrate the efficiency of the novel design of *Comet*. For more ablation experiments, we refer readers to the Appendix H.

We compare our methods on the model performance among PUMA (Dong et al. (2023)), Bumblebee (Lu et al. (2023)), Secformer (Luo et al. (2024)) and BOLT (Pang et al. (2024)), as they all using polynomial approximation for non-linear functions without fine-tuning. Our method demonstrates competitive model performance when compared to PUMA and BumbleBee, as in Table.6. For fair

Table 5: The communication (GB) and inference time (Second) comparison on Bert-base and RoBERTa-base model with Iron.

| Bert-base | Total Time | LayerNorm Time | LayerNorm Comm | Act Time | Act Comm | Softmax Time | Softmax Comm |
|---|---|---|---|---|---|---|---|
| Iron | 289.6 | 48.2 | 7.45 | 134.3 | 14.9 | 82.6 | 8.85 |
| *Comet* | **82.4(3.48×)** | **15.3(3.17×)** | **3.03(2.4×)** | **26.8(4.9×)** | **4.05(3.7×)** | **20.4(4.0×)** | **3.56(2.5×)** |
| RoBERTa | | | | | | | |
| Iron | 290.1 | 52.3 | 7.8 | 127.9 | 13.83 | 77.6 | 8.03 |
| *Comet* | **92.1(3.15×)** | **14.2(3.68×)** | **2.8(2.8×)** | **28.5(4.45×)** | **4.3(3.2×)** | **18.9(4.12×)** | **3.28(2.4×)** |

comparison, we also compare our method (integrated with other baselines) with original methods in inference time in Table. 7 in 2-PC mode. We exclude PUMA as it is in 3-PC computation. The Table. 7 shows our method achieve up to 3.2× speedup when compare to the state-of-the-are works.

Table 6: The model performance comparison of Comet, PUMA, and Bumblebee.

| | STS-B | CoLA | RTE |
|---|---|---|---|
| PUMA | 88.4 | 59.2 | 70 |
| Bumblebee | 87.5 | 60.8 | 70.04 |
| ours | 88.8 | 59.4 | 71.3 |

Table 7: The latency (communication (GB)) comparison between Bumblebee, BOLT, Secformer and Comet.

| | BOLT | Bumblebee | Secformer | Ours | Speedup |
|---|---|---|---|---|---|
| (14 calls) GeLU | 27.8s | 28.9s | 30.4s | 13.4s | 2.2× |
| Bert-base | 187.1s(60.61) | 153.4s(51.1) | 142.8s(63.4) | 57.9s(38.2) | 3.2× |
| Bert-large | 374.1s(93.3) | 303.3s(78.3) | 317s(82.3) | 234.5s(67.2) | 2.4× |

We also surpass MPCFormer in model performance and avoiding its lengthy knowledge distillation process, with similar inference time performance, as shown in Table. 8.

Table 8: The accuracy, and inference time (Second) comparison on Bert-base model with MPCFormer (Li et al. (2022)). "-" denotes not applicable to the method.

| Bert-base | Total Time | KD training | STS-B | CoLA |
|---|---|---|---|---|
| MPCFormer | 27.7 | 100 | 80.1 | 52.6 |
| *Comet* | 29.8 | - | 87.6 | 57.9 |

## 5 CONCLUSION

In this paper, we propose *Comet*, a communication-efficient and performant approximation framework for private Transformer inference. We specifically unified the hybrid complex protocols into one protocol– inverse square root for non-linear functions. Then, we further carefully design the double approximation method to convert the heavy communication of finding initial approximation to local computation for inverse square root, with our share flooding technique to securely secret sharing under strong assumption satisfaction. Our experimental results show that *Comet* outperforms prior art with up to 3.9× less communication and 3.48× speedups.

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

# APPENDIX

## A PRELIMINARY EXPERIMENTS ON ACTIVATION VALUE

We test the activation values distribution of Bert base model in various layers and different dataset, as we trained in Sec. 4.2. The Fig. 4 and Fig. 5 show that the activation values are compact to small absolute magnitude (surround zero) regardless the datasets and layers.

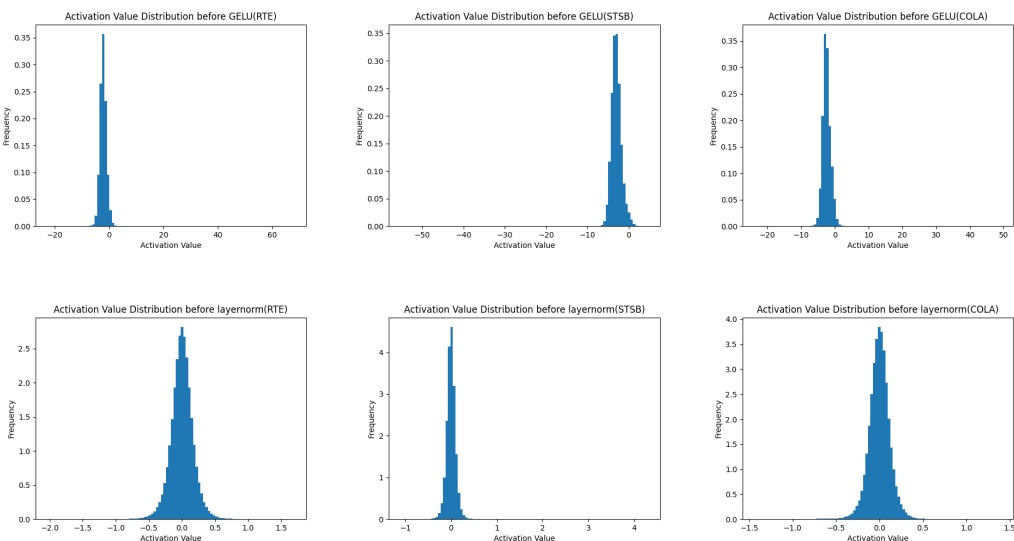

Figure 4: Activation distribution before GeLU and LayerNorm layer on various dataset (RTE, STS-B, COLA) within feed-forward block 2 of Bert base model

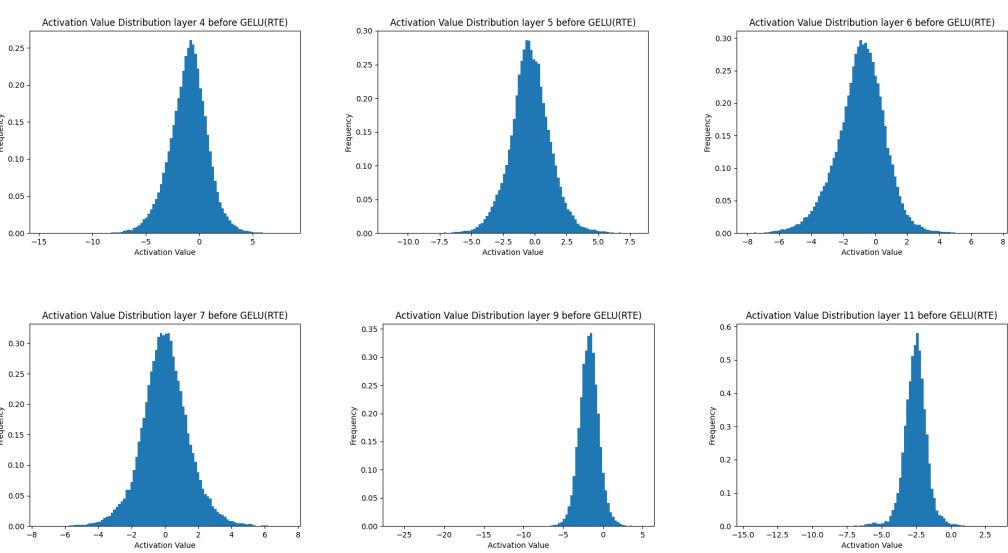

Figure 5: Activation distribution before GeLU layer on RTE in different block of Bert base model

## B    DERIVATION OF SMU FUNCTION

The SMU smoothed activation function that is proposed in  Biswas et al. (2022) is derived from the equation 11.

$$max(x_1, x_2) = \begin{cases} x_2 & \text{if } x_1 \leq x_2 \\ x_1 & \text{otherwise} \end{cases}$$
$$= \frac{(x_1 + x_2) + |x_1 + x_2|}{2} \tag{11}$$

By replacing $|x|$ with $\sqrt{x^2 + \mu^2}$, the smoothed approximation formula can be generated as shown in equation 12.

$$f(x_1, x_2, \mu) = \frac{(x_1 + x_2) + \sqrt{x^2 + \mu^2}}{2} \tag{12}$$

To have a smoothed approximation of Parametric Activation function, e.g., Leaky ReLU Xu et al. (2020), $x_1 = x$ and $x_2 = \alpha x$ are substituted and result in equation 13.

$$f(x_1, x_2, \alpha, \mu) = \frac{(1 + \alpha)x + \sqrt{(1 + \alpha)x^2 + \mu^2}}{2} \tag{13}$$

For convenience of calculation of inverse square root, we transform it into equation 14.

$$f(x, \alpha, \mu) = \frac{1 + \alpha}{2} * x + \frac{(1 - \alpha) * x^2 + \mu^2}{2 * \sqrt{(1 - \alpha) * x^2 + \mu^2}} \tag{14}$$

## C    EXPERIMENTS ON FLEXIBLE SMU TRAINABLE PARAMETERS

We test the model performance with flexible parameters in SMU function to replace the GeLU and ReLU in $Softmax^*$ on subset of GLUE benchmark. We follow the training setting and strategy as in Sec. 4.1 and Sec. 4.2. The Table. 9 shows about $1 - 2\%$ performance boost between unified model (s-GeLU + s-$Softmax^*$) and original one (GeLU + Softmax) on the datasets we test when using flexible trainable parameters.

Table 9: The model performance of a subset of GLUE benchmark with different combination of smoothed maximum unit replacement for GeLU and Softmax function. "$Softmax^*$" stands for ReLU replaced Softmax in Sec. 3.1. "s-" stands for the $smu(x)$ smoothed function in Sec. 3.1. Average Pearson and Spearman correlation is reported for STS-B. Matthews correlation is reported for CoLA. Accuracy is reported for other datasets.

| Bert-base | RTE | MRPC | STS-B | SST-2 | CoLA |
|---|---|---|---|---|---|
| GeLU + Softmax | 70.8 | 88.97 | 88.6 | 92.7 | 58.9 |
| GeLU + s-$Softmax^*$ | 68.2 | 87.1 | 86.9 | 91.1 | 56.6 |
| s-GeLU + Softmax | 70.4 | 86.7 | 88.7 | 91.5 | 56.8 |
| s-GeLU + $Softmax^*$ | 69.4 | 88.9 | 88.6 | 92.8 | 56.5 |
| s-GeLU + s-$Softmax^*$ | 72.1 | 89.5 | 89.2 | 93.8 | 59.4 |
| RoBERTa-base | | | | | |
| GeLU + Softmax | 74.2 | 92.3 | 89.1 | 92.1 | 55.7 |
| GeLU + s-$Softmax^*$ | 73.4 | 88.9 | 87.9 | 89.1 | 53.9 |
| s-GeLU + Softmax | 72.9 | 89.1 | 89.3 | 88.7 | 55.2 |
| s-GeLU + $Softmax^*$ | 73.7 | 91.9 | 89.2 | 88.4 | 55.8 |
| s-GeLU + s-$Softmax^*$ | 76.1 | 93.6 | 91.1 | 93.8 | 57.9 |

## D    CLOSENESS BOUND OF TWO SHARES EXPONENT

We evaluate the closeness bound of exponent parts between two shares to avoid the divergence of Newton method's iterations and keep the model performance. As the Fig. 6 shows, the upper bound

of closeness between shares in exponent is $|5|$, showing as "elbow point", to satisfy the second approximation assumption and result in good initial approximation without divergence.

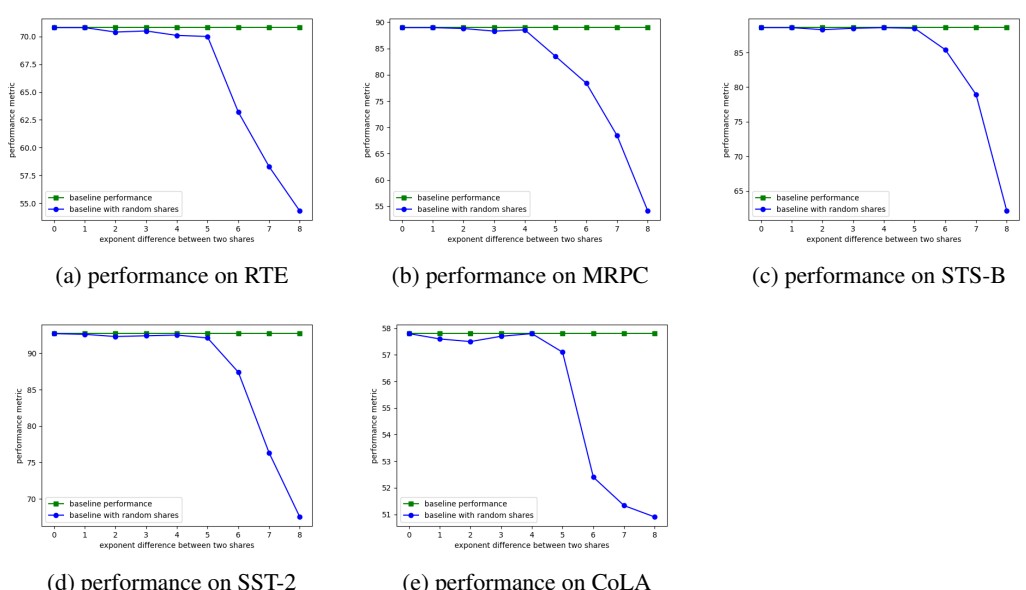

(a) performance on RTE      (b) performance on MRPC      (c) performance on STS-B

(d) performance on SST-2      (e) performance on CoLA

Figure 6: Closeness evaluation on Bert-base model with 4 Newton iterations on GLUE subset benchmark. "Baseline performance" denotes the original Bert-base model performance. "baseline with random shares" denotes the double approximation method with different closeness on exponent part of two shares.

## E  NEWTON METHOD'S LOCAL CONVERGENCE

We give the proof of local convergence of Newton's Method as following.

**Theorem 1.** *(local convergence of Newton's method) Let f be a twice continuously differentiable function defined over $\mathbb{R}^d$. Assume that (1) there exists a neighborhood $N_\sigma(x_*)$ of root of function $x_*$ and $M > 0$ for which $\|\nabla^2 f(x) - \nabla^2 f(y)\| \leq \frac{M}{2}\|x - y\|^2$ for any $x, y \in N_\sigma(x^*)$.*

*Proof.* we have

$$
\begin{aligned}
x_{k+1} - x_k &= x_k - \nabla^2 f(x_k)^{-1}\nabla f(x_k) - x_* \\
&= x_k - x_* + \nabla^2 f(x_k)^{-1}(\nabla f(x_k) - \nabla f(x_*)) \\
&= x_k - x_* + \nabla^2 f(x_k)^{-1}\int_0^1 [\nabla^2 f(x_k + t(x_* - x_k))](x_* - x_k)dt \\
&= [\nabla^2 f(x_k)]^{-1}\int_0^1 [\nabla^2 f(x_k + t(x_* - x_k)) - \nabla^2 f(x_k)](x_* - x_k)dt
\end{aligned}
$$

Then,

$$\|x_{k+1} - x_k\| \le \|[\nabla^2 f(x_k)]^{-1}\| \| \int_0^1 [\nabla^2 f(x_k + t(x_* - x_k)) - \nabla^2 f(x_k)](x_* - x_k)dt\|$$

$$\le \|[\nabla^2 f(x_k)]^{-1}\| \int_0^1 \|[\nabla^2 f(x_k + t(x_* - x_k)) - \nabla^2 f(x_k)](x_* - x_k)\|dt$$

$$\le \|[\nabla^2 f(x_k)]^{-1}\| \int_0^1 \|\nabla^2 f(x_k + t(x_* - x_k)) - \nabla^2 f(x_k)\|\|x_* - x_k\|dt$$

$$\le \int_0^1 Mt\|x_k - x_*\|^2 dt$$

$$\le \frac{M}{2}\|x_k - x_*\|^2$$

$\square$

## F  SMU Function Explanation

Although our proposed function, $\frac{1+\alpha}{2}x + 1/2 * \frac{(1-\alpha)x^2 + \mu^2}{\sqrt{(1-\alpha)x^2 + \mu^2}}$, can be simplified to $\frac{1+\alpha}{2}x + 1/2 * \sqrt{(1-\alpha)x^2 + \mu^2}$, when switch to such square root version, our efficient initial approximation finding method still works as our method is applicable to $f(x) = x^{1/a}$ ($a \in \mathcal{Z}_{\ne -1,0,1}$) functions, which is of independent interest. However, the Newton-Raphson update formula for square root, $y_{n+1} = 1/2 * (y_n + \frac{x}{y_n})$ is inefficient to compute on secret shares mode (Knott et al. (2021)). It requires secret share division that needs truncation and extension protocol on given integer ring (Rathee et al. (2020)).

Therefore, we switch the smu function to the inverse square root version. Since the Newton-Raphson update formula for inverse square root, $y_{n+1} = \frac{1}{2} * y_n(3 - xy_n^2)$ only requires efficient share multiplications and the inverse square root one is mathematically equal to the square root version. Our method contributes to a more lightweight initial approximation finding for later Newton iterations.

## G  Share Conversion

For the conversion between the floating point share in non-linear function and integer shares, a floating-point share ($= M + EL$) is correspond to one input value, according to IEEE 754 protocol. Such input value can be converted to the fixed-point share from decimal value to binary. After multiplied with a pre-defined scale ($= 2^f$), the $l$-bit fixed-point integer share can be obtained in the given integer ring $\mathcal{Z}_{2^l}$ (Knott et al. (2021)). We give the following share conversion example of 7.25:

$$\text{(float-point share)}0|10000001|1101000000000000000000(=1088946176(=M+EL)decimal) \longleftrightarrow$$

$$7.25 \longleftrightarrow \text{(fixed point)}0111.01^{(*2^4)} \longleftrightarrow_{(\div 2^4)} \text{(fixed point integer share)}01110100(= 29 \ decimal) \tag{15}$$

## H  Ablation Study

To demonstrate the challenge and show the effectiveness of the proposed share flooding, we conduct the ablation experiment on our method with and without flooding technique. The table 10 shows that the model performance can be significantly harmed without share flooding by 10 loss in corresponding metric. Our technique can successfully avoid such model performance loss while keeping the privacy.

Table 10: The ablation experiment of Comet method with v.s. without flooding technique.

|  | STS-B | CoLA | RTE |
| --- | --- | --- | --- |
| Comet without flooding | 72.2 | 49.7 | 62.5 |
| Comet | 87.9 | 57.9 | 71.1 |

