# OpenReview forum: "Comet: A Communication-efficient and Performant Approximation for Private Transformer Inference"
_ICLR.cc/2025/Conference — Submitted to ICLR 2025_

### Official Review · Reviewer_XSGo · 2024-10-27

**Soundness:** 3
**Presentation:** 3
**Contribution:** 3
**Rating:** 5
**Confidence:** 3

**Summary:**

This paper proposes a secure Multi-Party Computing (MPC)-based private inference framework, called Comet. Comet aims to address the challenge of high communication overhead, particularly for non-linear computations in MPC-based Transformer model inferences. Existing solutions, such as Look-Up Table (LUT) methods and aggressive polynomial approximations, often result in either high communication costs or reduced model performance. To tackle this, Comet harmonizes non-linear functions like GeLU and Softmax into a single approximation function, eliminating the need for heavy communication in finding initial approximations for Newton iterations. A “share flooding” technique is also proposed to further optimize secure two-party computations. Comet was evaluated on encoder-based Transformer models like BERT and RoBERTa (as well as on the evaluation benchmark GLUE) to demonstrate that it is both communication-efficient (i.e., fast in inference) and more accurate over the benchmark compared to existing MPC-based private Transformer inference frameworks.

**Strengths:**

- This paper is well-motivated. Studying methods that enable private inference over large language models has great practical potential for privacy-constrained use cases, such as in financial and medical institutions. Private inference methods (e.g., MPC-based approaches) can serve as potential alternatives to pure local computing.
- The paper is well-written, and the techniques proposed in the Comet framework appear sound.
- The experimental results look promising, especially regarding accuracy on the GLUE benchmark.

**Weaknesses:**

- The paper was motivated by using ChatGPT as an example, but all experiments were conducted on encoder-based Transformer models like BERT and RoBERTa. I understand that, architecturally, encoder models and decoder models differ mainly in language masking. However, models like GPT are autoregressive, which involves completely different computational intensity. For instance, autoregressive models generate tokens one by one, making it very difficult to batch inputs.
- Although MPC-based frameworks ensure that neither the client nor the server discloses information to each other, in Comet’s setting, the private information includes both the model weights and the users’ data. However, both parties still see the same (non-encrypted) output in these frameworks. While this may be acceptable for classification tasks conducted by BERT, it may not be suitable for generative models like ChatGPT (e.g., generated content could contain private user information).
- The absolute inference speed remains quite slow, e.g., around 30 seconds per batch, as shown in Table 3. Speeds like this make the proposed framework less practical.

**Questions:**

- As discussed in the “Weaknesses” section, there is concern about whether the proposed Comet framework can be applied to generative language models. Can the authors provide a feasible approach to extend the proposed framework to generative LLMs like ChatGPT and Llama?
- What if, in a practical application, the server model is not open-source (e.g., users can only call their API, like GPT and Claude)? Is there a way to still conduct private inference?
- The experiments in this paper are conducted on CPUs. However, Crypten supports GPU computing [1]. I wonder how the performance of Comet compares to Crypten on GPUs.
- It seems the major weakness of MPCFormer [2] is that it requires fine-tuning using knowledge distillation (KD). However, the idea makes sense for pure inference. I wonder if Comet could also benefit from KD.
- Encoder models are generally easy to batch. I wonder how different batch sizes affect the computation speed of Comet.

[1] https://github.com/facebookresearch/CrypTen?tab=readme-ov-file#installing-crypten
[2] https://openreview.net/forum?id=CWmvjOEhgH-

---

> ### Author Response · Authors · 2024-11-18
>
> We thank the reviews and questions from the reviewer. Here is the respones.
>
> >As discussed in the “Weaknesses” section, there is concern about whether the proposed Comet framework can be applied to generative language models. Can the authors provide a feasible approach to extend the proposed framework to generative LLMs like ChatGPT and Llama?
>
> As reviewer mentioned, the decoder-based model, like ChatGPT, maintains the same modules such as attention blocks and linear layers, so long as the non-linear functions, only adding masking when compared with encoder-based model like Bert. Thus, our method can also applied to such model as we focus on the functions rather than blocks. After the final non-linear layer (layernorm) calculation with our method, each party holds their share that corresponds to the final token vector, not the non-encrypted result. Therefore, the privacy for both parties is kept. For final word generation, the client can further encrypted its share to the server. Then, the server can infer the generated word in HE ciphertext and send back to client for decryption.
>
> >What if, in a practical application, the server model is not open-source (e.g., users can only call their API, like GPT and Claude)? Is there a way to still conduct private inference?
>
> Our two-party setting assumes the server model remain close-sourced and not disclosed with client. The client send encrypted input to server for calculation and only needs to send ciphertexts or secret shares to server during the whole inference, and vice versa. We find the example in question a bit vague and try to answer: if the API involves client and server communication, our method can conduct private inference. Our setting allows server keeps model, and client sends their input in HE ciphertext.
>
> We are happy to further discuss with reviewer if have further question.
>
> >The experiments in this paper are conducted on CPUs. However, Crypten supports GPU computing [1]. I wonder how the performance of Comet compares to Crypten on GPUs.
>
> We follow previous mainstream approaches for CPU-based privacy-preserving Transformer inference. The GPU support of CrypTen, which is CryptGPU[1], uses 2-out-of-3 Replicated Secret Shares in traditional 3 party. Such setting is not align with our work. Unfortunately, our work currently is not GPU-supported. We appreciate the reviewer's comment and would explore the support of our work on GPU computation in the future work.
>
> [1]S. Tan, B. Knott, Y. Tian and D. J. Wu, "CryptGPU: Fast Privacy-Preserving Machine Learning on the GPU," 2021 IEEE Symposium on Security and Privacy (SP),
>
> >It seems the major weakness of MPCFormer [2] is that it requires fine-tuning using knowledge distillation (KD). However, the idea makes sense for pure inference. I wonder if Comet could also benefit from KD.
>
> As our manuscript stated and the table 1 shows, our method does not need KD approach, as our method provide accurate results, and achieve high model performance when compared to MPCFormer, as the MPCFormer's simple 2-degree polynomial to replace softmax and GeLU functions leads to accuracy drop. We conduct an ablation experiment on our method with KD v.s. our method without KD in 10 epochs using the same setting in our manuscript, as shown in Table 2. Our method leverage KD can hardly further improve as it achieves similar performance as the teacher model.
>
> Table 1: accuracy comparison between MPCFormer, which using 100 epochs of KD, and Comet
>
> | Table 1   | KD(epochs) | STS-B | CoLA |
> | --------- | ---------- | ----- | ---- |
> | MPCFormer | 100        | 80.1  | 52.6 |
> | ours      | 0          | 87.6  | 57.9 |
>
> | Table 2   | RTE  | STS-B | CoLA |
> | --------- | ---- | ----- | ---- |
> | ours + KD  | 70.2     |   87.7    |  56.3    |
> | ours - KD | 71.5 | 87.6  | 57.9 |
>
> >Encoder models are generally easy to batch. I wonder how different batch sizes affect the computation speed of Comet.
>
> We tested our method in 64 input tokens and 128 tokens for Bert-based model, as following table. Our inference time shows roughly proportional relation to the input token batch.
> |      | 64 tokens | 128 tokens |
> |------|-----------|------------|
> | ours | 81.4s      | 158.3s     |
>
>
> >The absolute inference speed remains quite slow, e.g., around 30 seconds per batch, as shown in Table 3. Speeds like this make the proposed framework less practical.
>
> We agree with reviewer on there is a large latency gap for practical use, so long as other SOTA works. Our method pushes the latency boundary to a new limit, compared with other SOTAs. Additionally, our method can provide an inspiration to a new scheme of local computation for complex non-linear functions. We hope such insights can benefits the whole research community.

---

> > ### Author Response · Authors · 2024-11-30
> >
> > Dear reviewer,
> >
> > Thanks for your efforts. We are looking forward to your response and willing to discuss with you if have further questions, like the feasibility of our work in decoder-based model. If our response addressed your concern, we kindly ask you to consider raising rating accordingly.

---

### Official Review · Reviewer_CsYV · 2024-11-02

**Soundness:** 3
**Presentation:** 2
**Contribution:** 2
**Rating:** 6
**Confidence:** 4

**Summary:**

The paper introduces Comet, a framework aimed at efficient, privacy-preserving inference for Transformer models. Comet achieves this by unifying complex non-linear functions, like GeLU and Softmax, under a single protocol based on inverse square root calculations. It introduces a "double approximation" method to eliminate the need for extensive communication when finding initial approximations, further improving communication efficiency with a technique called "share flooding." Experimental results show that Comet reduces communication costs while maintaining good performance on models like BERT and RoBERTa on the GLUE benchmark

**Strengths:**

1. Designing an improved approximation for non-linear functions in private Transformer inference is critical, and this paper's adoption of the "smu" function is both innovative and effective, demonstrating strong accuracy after fine-tuning.
2. The paper presents an effective solution for obtaining a good initial approximation in the context of 2PC-based secret sharing, which is a key contribution to reducing communication overhead.

**Weaknesses:**

1. The properties of the "smu" function are not sufficiently explained. While experiments indicate that it achieves 1-2% higher accuracy than the original GeLU and Softmax functions, the underlying mechanism remains unclear. Including a graph of the approximated function would help clarify this.
2. The explanation of the double approximation method is confusing. Since MPC typically operates on fixed-point numbers, it is unclear whether the double approximation method is designed for floating-point operations, which are costly in MPC.
3. It is worth noting that other approximation methods like Bolt, Bumblebee, and Puma do not require fine-tuning, and this difference should be mentioned for context.

**Questions:**

Do you have code to reproduce your work since the "smu" function is rarely used?

---

> ### Author Response · Authors · 2024-11-16
>
> We thank the reviewer's positive rating. Here is the response.
> >The properties of the "smu" function are not sufficiently explained. While experiments indicate that it achieves 1-2% higher accuracy than the original GeLU and Softmax functions, the underlying mechanism remains unclear. Including a graph of the approximated function would help clarify this.
>
> We thank the suggestion for adding graph of the approximated function. Due to limited space, we refer reviewer to the Figure.1,2,3 in SMU (Biswas et al., CVPR'22) for detailed approximated function plot.
>
> >The explanation of the double approximation method is confusing. Since MPC typically operates on fixed-point numbers, it is unclear whether the double approximation method is designed for floating-point operations, which are costly in MPC.
>
> We are sorry for the cunfusion. During inference, we apply $M_{x_c}+LE_{x_c}$ and $M_{x_s}+LE_{x_{c|s}}$ fixed-point integer shares in Eq 10. The content previous is for explaining how it arrives at Eq 10. For detailed share conversion, please refer to Appendix G.
>
>
> >It is worth noting that other approximation methods like Bolt, Bumblebee, and Puma do not require fine-tuning, and this difference should be mentioned for context.
>
> We thank the suggestions and will indicate that in the accepted version (updated in blue font in latest revision).
>
> >Do you have code to reproduce your work since the "smu" function is rarely used?
>
> We would release the implementation regarding the accpetance. For SMU function implementation, we refer reviewer to https://github.com/iFe1er/SMU_pytorch

---

> > ### Comment · Reviewer_CsYV · 2024-11-27
> >
> > Thanks, my concerns are mostly addressed and I will keep my score.

---

### Official Review · Reviewer_W8wo · 2024-11-03

**Soundness:** 2
**Presentation:** 2
**Contribution:** 2
**Rating:** 6
**Confidence:** 5

**Summary:**

This paper introduces Comet to tackle to slow private inference for transformer based model. It is a plug-and-play method with an efficient approximation method to reduce the cost in non-linear computation. It achieves 3.9x less communication and 3.5x speed up compared to previous SOTA.

**Strengths:**

1. The methods are well motivated by observations.
2. The approximations are quite novel and interesting.

**Weaknesses:**

1. Figures are (in my opinion, not a major weakness) not good enough: (1) Figure 1 seems like hand written, with small fonts and large empty spaces. (2) Figure 2 should be put in appendix or wrapfigure (since they are not the main point of the paper).
2. The 2-relu approximations have been proposed in literature, e.g. MPCFormer. Also, since approximation has been heavily studied, can the author possible make a small table to present what is in literature and what is not (to improve presentation)?

**Questions:**

Please see the weakness section.

One other comment I have is that the experiment testbed is quite outdated. The author should consider more benchmark and larger scale model (This is out-of scope of the paper I think, but would be useful for future research), e.g. Vicuna and MMLU. ChatGPT is not Bert-like models.

---

> ### Author Response · Authors · 2024-11-16
>
> We thank the reviewer's positive rating. Here is the response.
> >Figures are (in my opinion, not a major weakness) not good enough: (1) Figure 1 seems like hand written, with small fonts and large empty spaces. (2) Figure 2 should be put in appendix or wrapfigure (since they are not the main point of the paper).
>
> We thank the reviewer's suggestion on figures and will improve accordingly in accepted version.
>
> >The 2-relu approximations have been proposed in literature, e.g. MPCFormer. Also, since approximation has been heavily studied, can the author possible make a small table to present what is in literature and what is not (to improve presentation)?
>
> We agree the reviewer's recommendations and will adding the table for better readability.
>
> >One other comment I have is that the experiment testbed is quite outdated. The author should consider more benchmark and larger scale model (This is out-of scope of the paper I think, but would be useful for future research), e.g. Vicuna and MMLU. ChatGPT is not Bert-like models.
>
> We thank the reviewer's suggestion. To make clear comparison with existing works, we follow the same benchmark and model choice. We will adding more test results for future research. We agree with ChatGPT is decoder-based model, with same functions as Bert. Our method can also applied to such decoder-based model.

---

> > ### Comment · Reviewer_W8wo · 2024-11-23
> >
> > Thank you for the reply. Will be happy to see the update.
> >
> > One other note, decoder-based models are different than Bert-based - Pre-filing and decoding are separate phrase that have different compute characteristics. For future research, you could try to see whether there is opportunity for approximation based method.

---

> ### Author Response · Authors · 2024-11-24
>
> Thank you for the response. We updated the Figure 1 with larger font and Figure 2 with wrapfigure, adding Table 1 for taxonomy of polynomail approximation. Please feel free to check the latest revision in blue font.
>
> We totally agree with your valuable suggestion on decoder-based models and its unique properties. We want to express our gratitude to your generous tip on future direction, and will explore this interesting area later on.
>
> We want thank again for your efforts in reviewing our work. Your positive rating or score raising will definately encourage our research with higher exposure to the community, potentially benefiting future works. We are always more than happy to answer your further questions.

---

### Official Review · Reviewer_HeTb · 2024-11-05

**Soundness:** 3
**Presentation:** 1
**Contribution:** 2
**Rating:** 5
**Confidence:** 4

**Summary:**

This paper tackles the communication (and latency) overhead introduced by nonlinear operations (GELU, LayerNorm, and Softmax) in cryptographically secure private inference of transformer-based models. The authors replace GELU and Softmax with a smoothed maximum unit (SMU) function, effectively transforming nonlinearities into inverse square root operations, and developed the methods such as double approximation and share flooding to efficiently compute them in privacy-preserving settings.

By mitigating the communication overhead associated with these nonlinear operations, this work tackles a major bottleneck for the practical deployment of private LLM services.

**Strengths:**

$\bullet$ Comprehensive evaluation of post-training performance across a variety of downstream tasks, demonstrating robustness, applicability, and generalizability of their proposed approaches.

$\bullet$ The integration of SMU with trainable parameters as an alternative to standard nonlinearities like GELU and ReLU-approximated Softmax is an interesting approach to simplifying the conventional nonlinearities and reducing communication and latency overheads in private inference.

$\bullet$ A thorough comparison with previous methods for private inference in transformer-based models.

**Weaknesses:**

$\bullet$ The contributions in this paper are largely engineering tweaks rather than substantive algorithmic advancements. While incorporating SMU (Biswas et al., CVPR'22) as a replacement for traditional nonlinearities like GELU and ReLU-approximated Softmax in a transformer-based model is an interesting approach, it lacks sufficient originality and novelty. Additionally, the use of Network Raphson methods for inverse square root calculations, as previously implemented in CryPTen but with an improved initialization, which reduces the number of iterations to converge, does not offer a particularly compelling approach. These techniques are often highly model- and task-dependent, requiring tuning when applied to other tasks or domains.

Furthermore, the authors’ proposed share flooding method, which is based on the observation  "that the absolute magnitude of tensor values is closely surrounded around zero (Line#317)," is rather intuitive and lacks deeper insight. Nonetheless, the authors will definitely get the points for the double approximation approach which effectively reduces the communication overhead to constant time complexity for lookup tables.

Overall, this paper lacks significant research insights or novel observations addressing the challenges of efficient private inference for large language models.

$\bullet$ I also find the comparison with CrypTen unclear, as CrypTen is a 3PC method, whereas the authors use a 2PC method. At one point (Line#482-483), the authors mention excluding PUMA due to its 3PC computation, which raises questions about the consistency of the comparison.

$\bullet$ The authors should consider comparing their approximation methods with the polynomial approximations of LayerNorm, GELU, and Softmax used in *Zimerman et al., Converting Transformers to Polynomial Form for Secure Inference Over Homomorphic Encryption* (**ICML'24**). It’s essential to at least qualitatively contrast and position the merits (efficiency and predictive performance) of their approach with these recent polynomial approximations.

$\bullet$ I find it quite challenging to follow the draft because the readability suffers from overly long paragraphs. Breaking up the content into shorter, clearer paragraphs would really help make the information easier to digest.

## Correction in the draft

$\bullet$ Figure 1 depicts the 2PC threat model for Post-LN configuration, however, in the caption the authors have mentioned CrypTen (which is 3PC). Also, in the FFN block diagram, one linear layer (after GELU layer) is missing.

$\bullet$ Line#180: GELU necessitates the Gaussian error function---> BERT and other recent transformer-based models adopt the $Tanh$ approximated GELU implementation, which is significantly faster, even in cryptographic settings (e.g., Bumblebee), than the original GELU   (`torch.nn.GELU()`). See [1] and their implementation in hugging face activation libraries [2]

1. https://paperswithcode.com/method/gelu


2. https://github.com/huggingface/transformers/blob/main/src/transformers/activations.py#L49

**Questions:**

$\bullet$ Why the authors have compared their 2PC method to 3PC method CrypTen? Is that the comparison with Crypten's Newton Raphson's method and other approximations, and not exactly with their 3PC settings?

$\bullet$ Why do the client and server share, which is an integer in the field arithmetic, converted to floating point numbers (Eq 1 and Eq 2)?

---

> ### Author Response · Authors · 2024-11-18
>
> >I find it quite challenging to follow the draft because the readability suffers from overly long paragraphs. Breaking up the content into shorter, clearer paragraphs ....
>
> We thank the suggestion on shorter paragraphs. We will try to make the accepted version multi-paragraph.
>
> >The contributions in this paper are largely engineering tweaks rather than substantive algorithmic advancements. this is an interesting approach, it lacks sufficient originality and novelty. Additionally, the use of Network Raphson methods for inverse square root calculations, as previously implemented in CryPTen but with an improved initialization, which reduces the number of iterations to converge, does not offer a particularly compelling approach. These techniques are often highly model- and task-dependent, requiring tuning when applied to other tasks or domains.
> Furthermore, the authors’ proposed share flooding method, which is based on the observation "that the absolute magnitude of tensor values is closely surrounded around zero," is rather intuitive and lacks deeper insight. Nonetheless, the authors will definitely get the points for the double approximation approach which effectively reduces the communication overhead to constant time complexity for lookup tables.
> Overall, this paper lacks significant research insights or novel observations addressing the challenges of efficient private inference.
>
>
> First, we argue that our work is not merely a set of "engineering tweaks". The SOTA works, like SiRNN and Iron, on calculating accurate results of the complex non-linear functions (e.g., inverse of square root) all requires pre-generated Look-Up Table to find an inital approximation, which introduce heavy communication in two-party settings. This fundamental issue is the latency bottleneck of secure Transformer inference. Our method solves this fundamental problem with carefully designed approximation that origin from the long-time overlooked property of secret share. This design achieves to remove the heavy look-up table with comparable results. In addition, our method can be applied to other general functions $f(x) = x^{1/a}$ where $a \in \mathcal{Z}_{\ne -1,0,1}$ for finding inital approximations. The inverse of square root that showed in secure Transformer inference is just one possible application.
>
> Second, we argue that our method is not task/model dependent, as our work focus on improving the efficiency of the non-linear functions like inverse square root. The inverse square root is a vital building block to many ML applications. Our method also has been extensively tested in many tasks without fine-tuning, as shown in Evaluation/Appendix part. For the reviewer's share flooding comment, we first thank the reviewer giving credits to our method's correctness. We argue that being intuitive does not mean lack deeper insight and it should not be a "sin" for a work. A work based on overlooked but solid observation should be more valued, as it could be easier-to-follow.
>
> Finally, we highly recommand reviewer re-evaluate our work with fair and objective comments. We are happy to read the further proof for "lack of novalty" with proper citation.
>
>
> >I also find the comparison with CrypTen unclear, as CrypTen is a 3PC method, whereas the authors use a 2PC method. At one point, the authors mention excluding PUMA due to its 3PC computation, which raises questions about the consistency of the comparison.
>
> CrypTen uses a trusted-thrid party for generating secret shares offline and uses 2-out-of-2 additive secret share for online inference, which aligns with our method setting for the online part. While PUMA uses 2-out-of-3 Replicated Secret share, which involves three parties for inference.
>
> >The authors should consider comparing their approximation methods with the polynomial approximations used in Zimerman et al., Converting Transformers to Polynomial Form for Secure Inference Over Homomorphic Encryption (ICML'24).
>
> The paper cited focus on proposing more stable HE-oriented models by retraining the existing Remez polynomial approximation with hyperparameters, which is orthogonal with our work. We also compare same type high-order Taylor expansion polynomial approximation as Remez poly, showing $\sim{4.3\times}$ speedup. We would consider comparing Remez poly approximaiton in accepted version. We hope reviewer evaluate our work based on the merits, rather only on completeness of experiments.
>
>
>
> >Correction in the questions
>
> - The tanh in private inference also applies look-up table for calculation, which is not significant faster than the erf approxi.
> - "Network" --> Newton in Q2.
>
> >Why do the client and server share, which is an integer in the field arithmetic, converted to floating point numbers (Eq 1 and Eq 2)?
>
> During inference, we apply $M_{x_c}+LE_{x_c}$ and $M_{x_s}+LE_{x_{c|s}}$ integer share in Eq 10. The content previous in Sec.3.2 is for explaining how it arrives at Eq 10. For share conversion, please refer to Appendix G.

---

> ### Comment · Reviewer_HeTb · 2024-11-22
> **Response to the Author's Rebuttal**
>
> Thanks for providing the detailed rebuttal.
>
> >We argue that being intuitive does not mean lack deeper insight and it should not be a "sin" for a work. A work based on overlooked but solid observation should be more valued, as it could be easier-to-follow.
>
> I completely agree with the author's sentiment on this (**theory often follows invention**). I believe that the authors have already presented empirical observations in Appendix A, on the concentration of activation values, which I overlooked in the first round of reviews, Thus, my initial comment: "is rather intuitive and lacks deeper insight" is resolved now.
>
> I agree that "novelty" is a very subjective term and largely depends on the reviewer's perspective. Also, the author should **not** be heavily penalized for overlooking the comparison with some recent work.
>
> Nonetheless, it is quite important to position the paper among the existing work, how the author's work complements other's research, and what additional perspective it brings to advance the field. I believe that the author's contribution to reducing the LUT complexity to constant time is valuable.
>
> I read the comments from the other reviewer and strongly believe that the paper will benefit from another round of review and the authors have received sufficient feedback to improve the paper. There are significant outstanding issues, especially regarding the clarity and justification of the author's method.
>
> Based on the re-assessment, I have increased the score to 5.

---

> > ### Author Response · Authors · 2024-11-22
> >
> > Dear reviewer,
> >
> > We are much appreciated about your responds and score raising. We are happy to answer your further questions or make any necessary changes to the maunscript for better clarity and justifications.

---

### Meta-Review · Area_Chair_8xwB · 2024-12-24

**Metareview:**

This paper tackles the communication (and latency) overhead introduced by nonlinear operations in secure transformer inference. The authors replaced GELU and Softmax with a smoothed maximum unit (SMU) function and developed double approximation and share flooding to efficiently compute them in privacy-preserving settings.

Strength of this paper:
- secure transformer inference is an important problem
- the share flooding technique sounds intuitive

Weakness of this paper:
- several key components of this paper (e.g., approximation of non-linear operations) were developed in previous papers, and this paper seems to simply borrow them into secure inference, hence the ideas are not very refreshing.
- The proposed framework and experiments conducted are mainly for transformer encoders (e.g., BERT), yet the authors used ChatGPT, which is a decoder models with autoregressive generation at inference, as the main driver for this paper. This seems not coherent.
- The absolute inference time even for BERT remained impractical

The weaknesses outweigh the strengths, hence a rejection.

**Additional Comments On Reviewer Discussion:**

The authors provided rebuttals to address some of the reviewer's questions, but two reviewers who provided very solid and detailed reviews remained negative.

---

### Decision · Program_Chairs · 2025-01-22

Reject